# Effects of Urban Park Environmental Factors on Landscape Preference Based on Spatiotemporal Distribution Characteristics of Visitors

**Mengwei Yang** [†] [ID]**, Renwu Wu** [†] [ID]**, Zhiyi Bao** [ID]**, Hai Yan** *[ID]**, Xinge Nan** [ID]**, Yixin Luo and Tingfang Dai** [ID]

School of Landscape Architecture, Zhejiang Agriculture and Forestry University, Hangzhou 311300, China; yangmw@stu.zafu.edu.cn (M.Y.); wurenwu0034@zafu.edu.cn (R.W.); 20070007@zafu.edu.cn (Z.B.); eminem@stu.zafu.edu.cn (X.N.); 2020605052045@stu.zafu.edu.cn (Y.L.); 2020605052013@stu.zafu.edu.cn (T.D.)

* Correspondence: yanhai@zafu.edu.cn
† These authors contributed equally to this work.

**Abstract:** Urban parks are public green spaces which have a direct impact on the daily outdoor activities of residents and visitors due to their landscape and functionality. To enhance the spatial vitality and services of urban parks to meet the needs of urban residents and visitors, managers and planners should focus on people's perceptions and preferences of park landscape characteristics. This study aimed to investigate the relationship between visitors' landscape preferences and environmental factors in urban parks. Fixed-point photography and mobile measurements were used to quantify the environmental factors of urban parks. Unmanned aerial vehicle observations and ground observations were used to examine physical activities and spatial behaviors of visitors to quantify their landscape preferences. Second, the differences in visitors' landscape preferences in various types of landscape spaces were analyzed based on descriptive statistics and significance tests. Finally, a correlation analysis and principal component analysis were introduced to explore the relationship between urban park environmental factors and visitors' landscape preferences. The results showed that visitors' physical activities and spatial behaviors were affected by many environmental factors, especially accessibility and visible green index. Our findings also shed new light on the significant differences in visitors' physical activities and spatial behaviors of different genders and age groups. It was observed that most male visitors were sitting, whereas female visitors preferred to take photographs. Exploring the relationship between urban parks and visitors' landscape preferences is of great significance for improving park satisfaction, people's happiness, and urban sustainability.

**Keywords:** urban parks; landscape preference; environmental factors; physical activity; spatial behavior; landscape design

## 1. Introduction

The rapid urbanization and the impact of the COVID-19 pandemic have placed various physical and mental pressures on urban residents. Many studies have shown that getting close to nature and taking part in outdoor leisure activities are good ways to relieve stress, which is of great significance to the health and well-being of residents. As a type of man-made nature in the city, urban parks provide a variety of ecosystem services for urban residents, such as alleviating the urban heat island effect, purifying air, and maintaining ecosystem diversity [1]. At the same time, urban parks provide recreational places for urban residents, which can enhance the communication between humans and nature, promote the physical and mental health of urban residents, relieve work pressure, enhance happiness, and maintain good social relations [2]. Many studies have shown that there is a significant positive correlation between urban parks and public health, which manifests in many aspects, such as physical, psychological, and social health [3]. To enhance the spatial vitality and services of urban parks to meet the needs of urban residents, managers and planners

should focus on people's perceptions and preferences of park landscape characteristics [4] and strive to create urban parks with aesthetic and functional qualities [5].

Stigsdotter conducted a survey and explored stressed individuals' preferences for activities and environmental characteristics [6]. The results of this study provide some indicators as to how urban green spaces can be planned and designed to satisfy the needs and preferences of stressed individuals. Hofmann explored the differences in preferences between landscape planners and urban residents by sorting and rating photographs of parks and urban derelict land, and found that preferences varied between groups [7]; landscape planners preferred natural areas with low accessibility and high species abundance, and residents showed a greater preference for formal parks. Some studies have also found that landscape preference reflects the effect of landscape on human attention recovery to some extent [8,9]. Understanding landscape preference helps identify what kind of landscape is most favored, based on the comprehensive evaluation of the preferences of individuals on the landscape [10,11]. Previous studies on landscape preferences have mainly focused on the influence of environmental factors. Natural environmental factors such as plants and water are significantly correlated with landscape preference [12]. For instance, an increase in plant species and plant density can significantly promote individual landscape preferences [13,14].

Regarding the methods, most landscape preference studies were performed by asking interviewees to rate the degree of beauty or preference of photos with different environmental factors. Simultaneously, experts were asked to score or objectively quantify environmental factors, and correlation analysis was then carried out to determine the relationship between environmental factors and landscape preference [15]. However, this approach needs to be strengthened in terms of scientifically controlling the environmental variables while excluding factors such as thermal comfort, which result in biased study results. In recent years, the photo simulation method has been used to modify a single environmental factor in the picture to effectively control these variables, with the aim of comparing the presence or absence of environmental factors and how these influence landscape preference [16]. Furthermore, only a few researchers have applied the photo simulation method to study the permutations and combinations of environmental factors. Most of these studies have only modified a small number of variables for each group of control photos, making it difficult to study changes in landscape preferences in the case of multiple environmental factors.

In practice, the public does not evaluate environmental factors individually but evaluates the combination of environmental factors, resulting in preferences. Among the research on multiple influencing factors of park landscape preference, most of the research focuses on visual scale. However, big data surveys show that in addition to visual factors, accessibility, thermal comfort, leisure facilities, and other factors will have an impact on tourists' landscape preferences [17]. Therefore, a systematic and comprehensive understanding of the relationship between visitor preferences and park environmental factors can provide management guidelines for landscape planners to optimize the landscape configuration of urban parks and promote greater integration of functional and practical urban parks into urban life.

The relationship between visitors' landscape preferences and environmental factors in urban parks is the focus of this study. It mainly includes the following two parts. (1) How do the different environmental factors of urban parks affect visitors' landscape preferences? (2) What are the major environmental factors of urban parks that affect visitors' landscape preferences? To answer these questions, an urban park in Hangzhou was selected as an example. Fixed-point photography and mobile measurements were used to quantify the environmental factors of urban parks. Unmanned aerial vehicle (UAV) observations and ground observations were used to examine the distribution of visitors to quantify their landscape preferences. Second, the differences in visitors' landscape preferences in various types of landscape spaces were analyzed based on descriptive statistics and significance tests. Finally, a correlation analysis and principal component

analysis (PCA) was used to investigate the environmental factors of urban parks that affect visitors' landscape preferences.

## 2. Materials and Methods

### 2.1. Study Area

Hangzhou (29°11–30°34′ N; 118°20′–120°37′ E), the capital of Zhejiang Province, is recognized throughout China as a garden city and is renowned for its tree-lined streets and scenic West Lake National Park. By the end of 2021, the population of permanent residents was 12.0 million, and per capita green area of the park was approximately 13.55 m². Hangzhou has a subtropical monsoonal climate. Annually the area experiences average temperatures of 17.8 °C, relative humidity of 70.3%, 1454 mm of precipitation, and 1765 h of sunshine. Summer is hot and humid, winter is cold and dry, and spring and autumn have pleasant weather and are the most suitable seasons for sightseeing.

Taiziwan Park, located in the southwestern corner of West Lake, was selected as a study area. The park has a vegetative landscape with romantic characteristics and an area of approximately 80.03 ha. These parks preserve traditional Chinese gardens and draw on the landscaping techniques of Western gardens. The spring landscape is a special feature of Taiziwan Park, where cherry blossoms and tulip flowers attract many visitors. Six sites were selected as study areas: Wangshan Lawn, Pipa Islet, Xiaoyao Hillside, Tianyuan Terrace, Rose Garden, and Zhulianbi Waterfall, including 24 sample points (Figure 1 and Table 1).

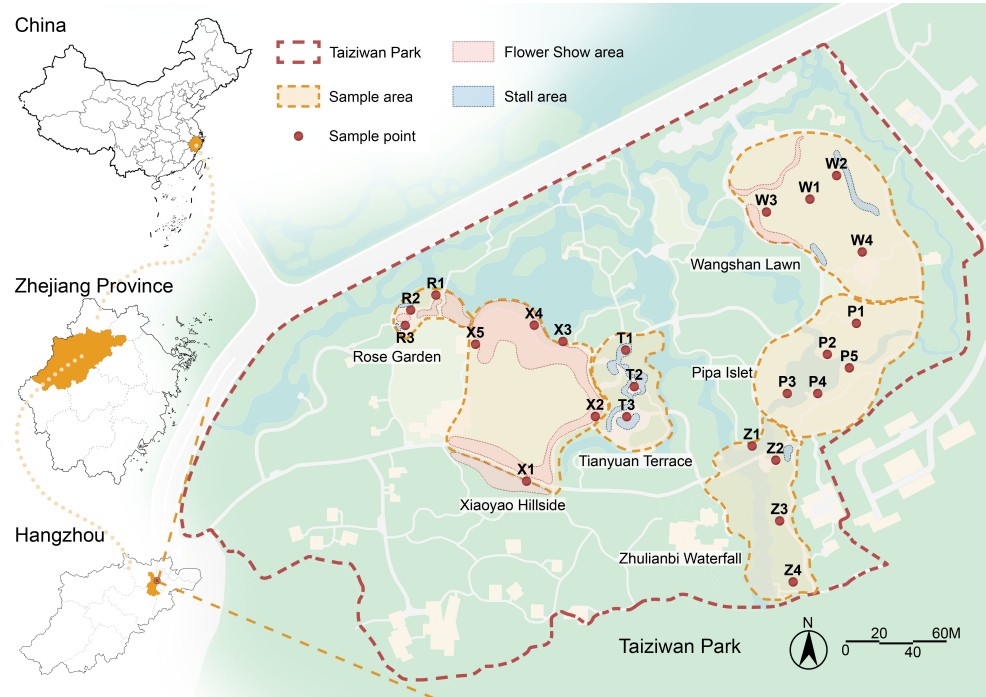

**Figure 1.** Location of six sample areas and 24 sample points in Taiziwan Park.

**Table 1.** Main characteristics of sample areas.

| Name | Number of Sample Points | Structure | Vegetation | Underlying Surface | Leisure Facilities |
|---|---|---|---|---|---|
| Rose Garden (R) | 3 | Sculpture | Flowers, trees | Wood platform | Seat |
| Pipa Islet (P) | 5 | Qingwan Pavilion | Lawns, trees | Lawn | Seat, lawn |

| Name | Number of Sample Points | Structure | Vegetation | Underlying Surface | Leisure Facilities |
|---|---|---|---|---|---|
| Tianyua Terrace (T) | 3 | Stalls | Trees | Brick paving | Seat |
| Wangshan Lawn (W) | 4 | Windmill | Lawn, trees | Lawn | Lawn |
| Xiaoyao Hillside (X) | 5 | Church | Flowers, lawn, trees | Lawn | Seat |
| Zhulianbi Waterfall (Z) | 4 | Shicui Pavilion | Trees | Stone paving | Seat |

*2.2. Quantifying Environmental Factors and Landscape Preferences*

Environmental indicators were used to quantify environmental factors, and these were divided into structural and vegetative indicators. Sixteen visual and physical environmental indicators were selected to explore the relationship between environmental factors and landscape preferences, providing a more comprehensive description of the environmental characteristics of each landscape space (Table 2). The selection of 16 environmental factors is based on the synthesis of research and multi-sensory literature. Through random interviews with park visitors and reference to relevant literature, 16 environmental factors affecting visitors' landscape preference are determined, and the main factors affecting visitors' landscape preference are explored. To quantify the environmental factors, panoramic images of the sample points were captured (measured using an iPhone and fixed height 1.65 m) (Figure 2). Environmental indicators, such as accessibility, visible green index (VGI), spatial openness, building, water, tree, shrub, herb, flowering plant, and foliage plant abundance, were measured based on the proportion of the area occupied by the factors in the panoramic images. The average of three panoramic images was measured for the spring conditions to reduce the uncertainty of the results (Figure 3). Environmental indicators, such as species, flowering plant, color richness, and adequacy of leisure facilities, were measured using the mean of panoramic images combined with field measurements. From March to April 2022, mobile measurements were used to record micro-climate data, such as air temperature, humidity, wind speed (measured using a TES-1365 humidity temperature meter and AS836 digital anemometer), and noise values of the sample points. To quantify sky view factor, fisheye images of the sample points were captured (measured using Canon EOS 6D Mark and Sigma 8mm Circular Fisheye Lens) (Figure 2). As a thermal indicator derived from the human energy balance, the physiologically equivalent temperature is well-suited for the evaluation of thermal comfort, which was calculated using the RayMan model [18]. For its calculation, micro-climate data and sky view factor were imported into the model (Age: 35 years, Weight: 75 kg, Height: 1.75 m, working metabolism: 80 W of light activity, and the heat resistance of clothing: 0.9 clo). Soundscape quality was quantified using noise values. Measurements on typical days with no wind or rain were performed to reduce the uncertainty of the results.

Spatiotemporal distribution characteristics were used to assess landscape preferences with the corresponding characteristics of the visitors, whereas the landscape preference indicator (*LPI*) was used to assess only the landscape preferences of the visitors. These were quantified by the number of visitors at each sample point. Furthermore, characteristic indicators of visitors, including the time of day (8:00–10:00, 10:00–12:00, 12:00–14:00, 14:00–16:00, 16:00–18:00), day of the week (weekdays and weekends), gender (male and female), age group (elderly, young, and children), and behavior, were used to describe differences between sample points. From March to April 2022, including weekdays and weekends, UAV and ground observations were used to record the characteristics of visitors. Visitors' age groups were estimated and categorized, and their behaviors were marked on park maps. According to the observations, visitors' age groups were divided into three types (the elderly were those over 60 years old, children were those under 15 years old,

and the rest were categorized as young), and their behaviors were divided into six types (sitting, photography, standing and chatting, walking, sporting and exercising, and children playing).

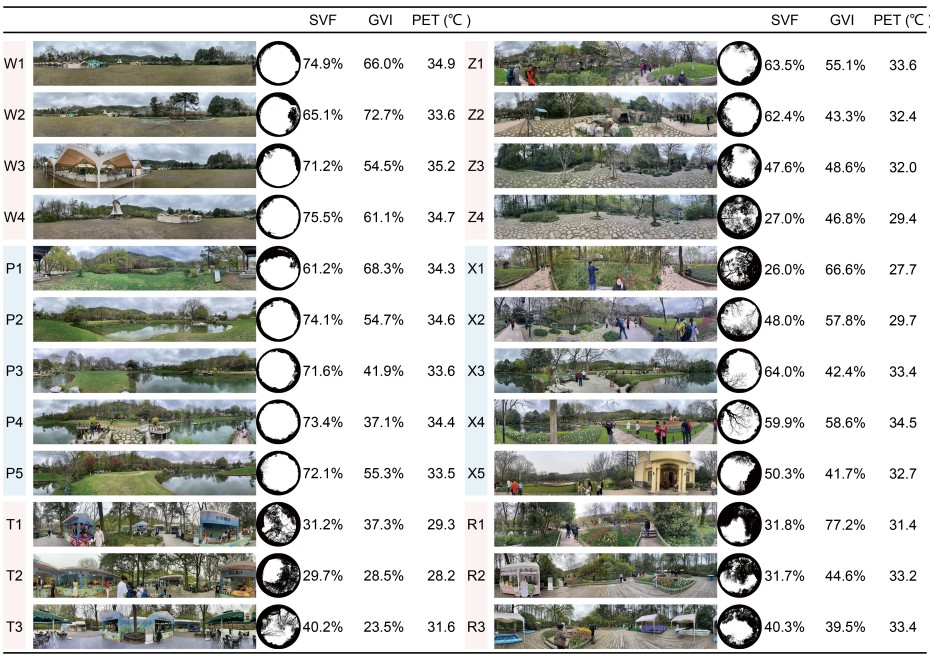

**Figure 2.** Environmental characteristics of 24 sample points (SVF: sky view factor; GVI: visible green index; PET: physiologically equivalent temperature).

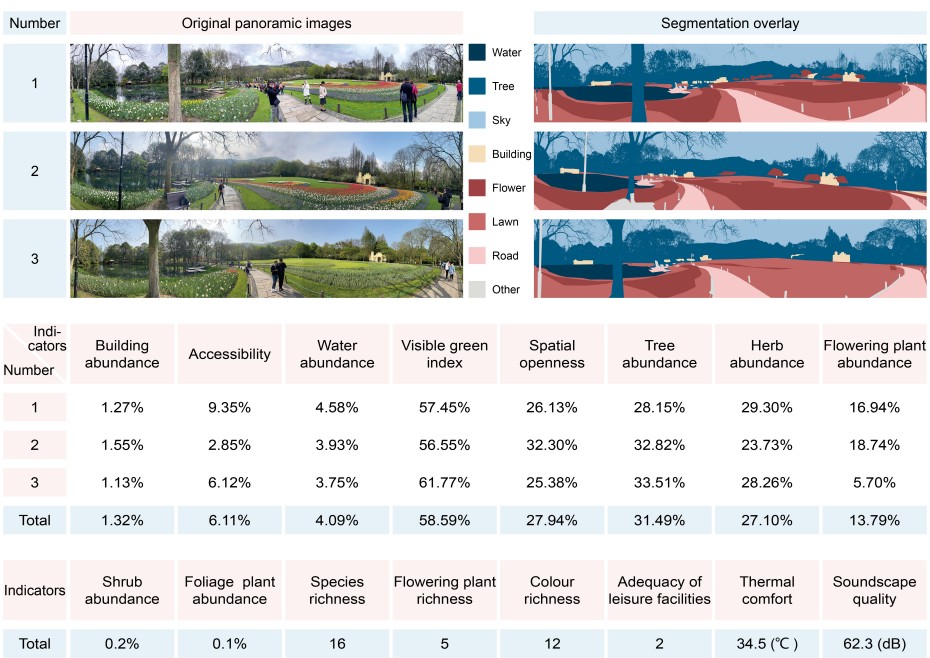

**Figure 3.** An example of quantifying the environmental factors of sample points.

**Table 2.** Environmental factors that potentially influence visitors' landscape preferences.

| Attribute | Indicator Number | Indicators | Description |
|---|---|---|---|
| Structural indicators | 1 | Building abundance | The percentage of building area in the overall area of the panoramic image. |
| | 2 | Accessibility | The percentage of road area in the overall area of the panoramic image. |
| | 3 | Water abundance | The percentage of water area in the overall area of the panoramic image. |
| | 4 | Visible green index | The percentage of vegetative area in the overall area of the panoramic image. |
| | 5 | Spatial openness | The percentage of sky area in the overall area of the panoramic image. |
| | 6 | Adequacy of leisure facilities | The number of seats in the panoramic images. |
| | 7 | Thermal comfort | Quantify by physiologically equivalent temperature (°C). |
| | 8 | Soundscape quality | Quantify by noise values (dB). |
| Vegetation indicators | 9 | Tree abundance | The percentage of tree area in the overall area of the panoramic image. |
| | 10 | Shrub abundance | The percentage of shrub area in the overall area of the panoramic image. |
| | 11 | Herb abundance | The percentage of herb area in the overall area of the panoramic image. |
| | 12 | Flowering plant abundance | The percentage of flowering plant area in the overall area of the panoramic image. |
| | 13 | Foliage plant abundance | The percentage of foliage plant area in the overall area of the panoramic image. |
| | 14 | Species richness | The number of plant species in the panoramic images. |
| | 15 | Flowering plant richness | The number of flowering plant species in the panoramic images. |
| | 16 | Color richness | The number of plant colors in the panoramic images. |

### 2.3. Evaluating Differences in Spatiotemporal Distribution Characteristics of Visitors

The differences in the spatiotemporal distribution characteristics of visitors in different landscape spaces were quantified using statistical analysis. The spatiotemporal distribution and behavioral characteristics of visitors in different landscape spaces are described and compared by descriptive statistics. Additionally, the study tested whether there were significant differences in visitors' landscape preferences in different landscape spaces. A difference significance test was chosen based on the data distribution. However, because the data of visitor characteristics did not conform to the normal distribution, a non-parametric significance test (Kruskal–Wallis test) was used in these studies.

### 2.4. Exploring Relationships between Environmental Indicators and Landscape Preferences

Correlation analysis and PCA were used to determine the impact of environmental factors on visitors' landscape preference. First, a Spearman correlation analysis was conducted to identify any significant correlations between environmental factors and landscape preference. Second, a PCA was performed using environmental indicators and landscape preference to obtain a nonlinear relationship between them, so as to explore the LPI of

different environmental factors. Finally, the importance of each indicator was ranked and a combined score for each sample point was calculated.

## 3. Results

### 3.1. Visitor Characteristics

Figure 4 shows the general characteristics of the visitors in Taiziwan Park during spring. The total number of visitors to Taiziwan Park was higher on weekends than on weekdays, accounting for more than 59.3% of all visitors. Visitors to the park were mainly female (65.3%), and mostly from the young age group (63.1%), followed by the elderly (23.0%) and children younger than 15 years of age (13.9%). The sum of number of visitors to the park peaked between 2:00 pm and 4:00 pm, with troughs occurring from 8:00 am to 10:00 am on both weekdays and weekends. Sitting (35.5%) and photography (31.5%) were the main activities, followed by standing and chatting (11.4%) and walking (9.4%), with children playing (9.0%) and sporting and exercising (3.2%) being the least frequent.

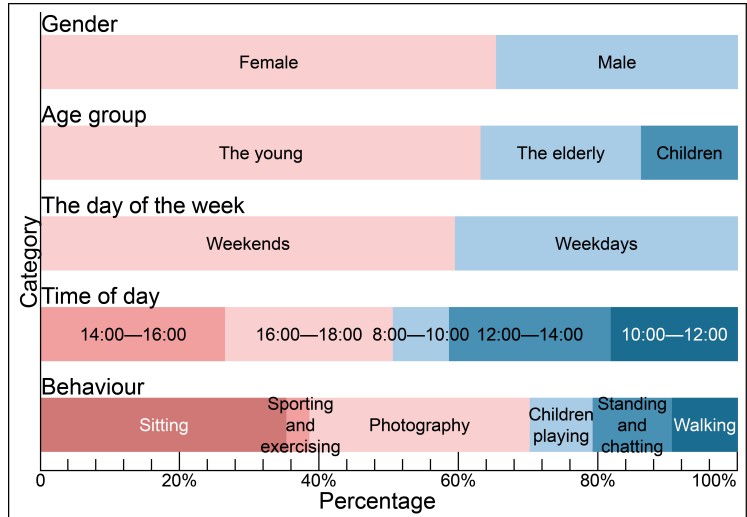

**Figure 4.** Descriptive statistics of the basic information of visitors.

According to the results of the non-parametric significance test, there were significant differences between the visitors in terms of the day of the week, time of day, sex, age group, and behavior (Figure 5). As shown in Figure 5a, males (65.1%) were more likely to visit the park on weekends than females (56.2%). The elderly (60.7%) were more likely to visit the park on weekdays than young people (38.9%) and children (16.0%). There were more children (19.7%) than elderly (15.2%) on weekends, in contrast to fewer children (5.5%) than elderly (34.2%) on weekdays. The primary behavior of visitors on weekdays was photography (37.6%), and sitting (38.3%) was most frequent on weekends. In addition, there were fewer male youths (59.2%) than female youths (65.2%) but more male children (19.6%) than female children (10.9%). Females (34.9%) showed a stronger preference for photography than males (25.1%). From a time perspective (Figure 5b), the proportion of children was highest at 12:00–14:00, while that of young people was highest at 14:00–16:00, and that of the elderly at 10:00–12:00 (children, 30.0%; the young, 29.0%; the elderly, 30.1%). The elderly were the main visitor groups present at 8:00–10:00, while young people were present at any other time of the day. Moreover, the main behavior of visitors was photography at 8:00–10:00, while sitting was most popular at any other time.

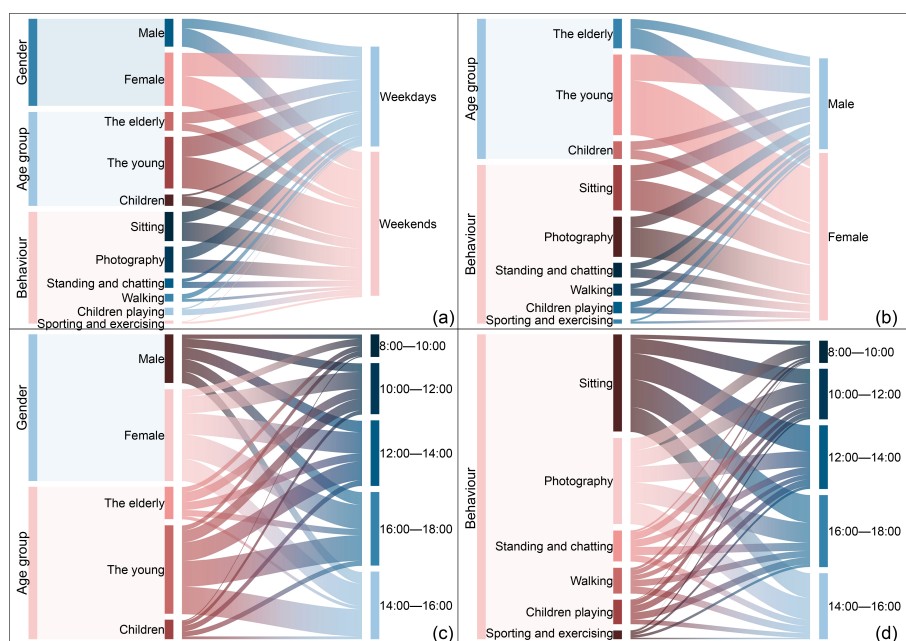

**Figure 5.** Comparison of visitor characteristics. (**a**) Differences between the day of the week. (**b**) Differences between gender. (**c**,**d**) Differences between time of day.

## 3.2. Visitors' Physical Activities and Spatial Behaviors

According to the data (Figure 6), Xiaoyao Hillside (21.1%) had the highest number of visitors in the sample areas, followed by Wangshan Lawn (19.6%) and Pipa Islet (17.6%). Additionally, of the sample points within sites, T3 (6.8%) had the highest number of visitors, followed by X5 (7.0%) and W2 (6.0%).

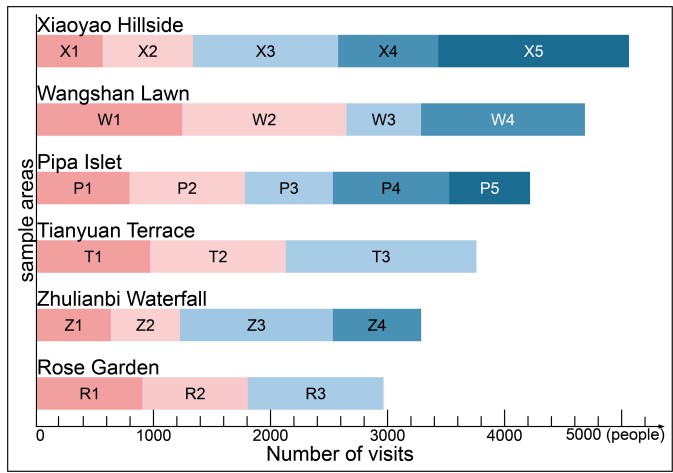

**Figure 6.** Number of visits to 24 sample points in the six sample areas.

Figure 7a shows that females (22.0%) and the elderly (26.9%) were primarily found in Xiaoyao Hillside for photography, males (20.2%) and children (24.5%) were mostly found in Wangshan Lawn for playing, and young people (21.0%) gathered in Xiaoyao Hillside and Wangshan Lawn for photography. In terms of time (Figure 7b), Xiaoyao Hillside had the highest number of visitors on weekdays (24.8%), and Wangshan Lawn had the highest number of visitors on weekends (20.2%). Furthermore, visitors were mainly distributed in Xiaoyao Hillside at 8:00–10:00 (30.1%), 10:00–12:00 (20.2%), and 16:00–18:00 (20.7%) and at Tianyuan Terrace (19.8%) at 12:00–14:00 and Wangshan Lawn (20.4%) at 14:00–16:00.

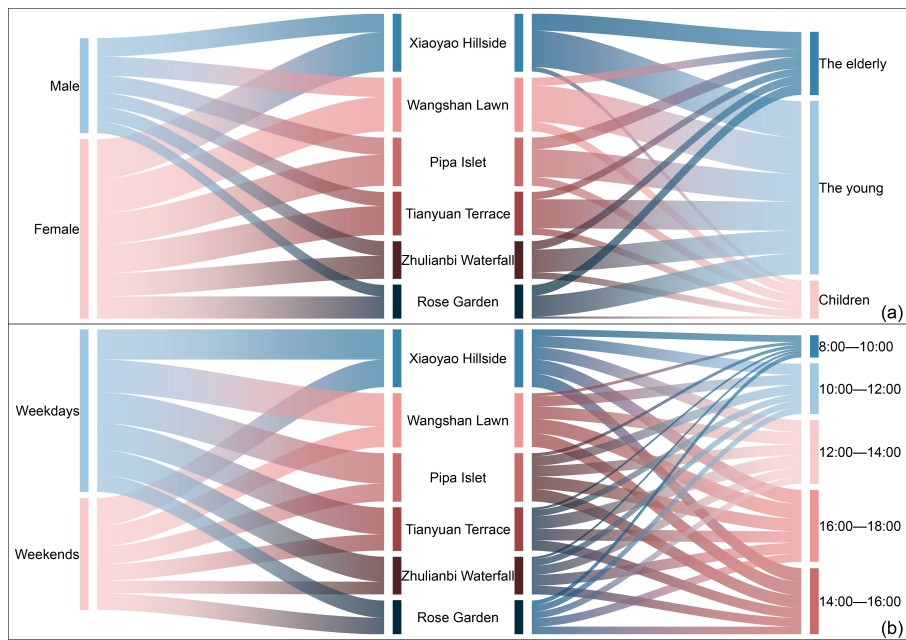

**Figure 7.** Comparison of visitors' physical activity and spatial behavior. (**a**) Differences between gender and age group in sample areas. (**b**) Differences between the day of the week and time of day in sample areas.

Visitors were mainly distributed in the Tianyuan Terrace (41.1%) for sitting, Wangshan Lawn for standing and chatting (30.8%), walking (36.2%), and children playing (39.2%), Zhulianbi Waterfall (86.8%) for sporting and exercising, and Xiaoyao Hillside (36.6%) for photography (Figure 8).

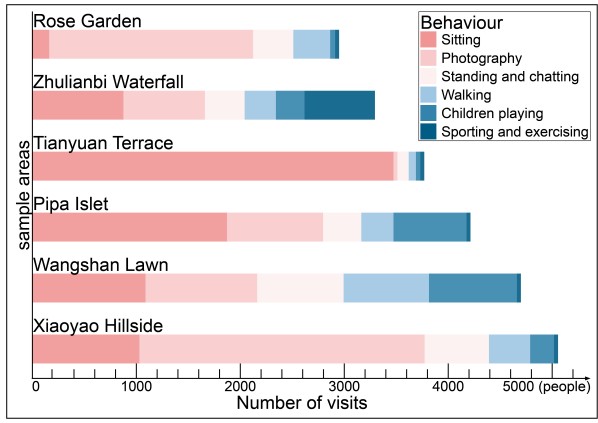

**Figure 8.** Comparison of behavior characteristics of visitors.

Figure 9a shows that males (6.6%) were mostly found in W4, whereas females (7.1%) were mostly found in T3. The elderly (7.9%) and young individuals (7.4%) were mostly found in X5, and children (9.0%) were mostly found in W1. In addition, visitors were mainly distributed in T3 (18.3%) for sitting, W4 (13.2%) for standing and chatting, W3 (15.9%) for walking, Z4 (60.3%) for sporting and exercising, X5 (15.5%) for photography, and W1 (15.9%) for children playing (Figure 9b). In terms of time, most visitors were in X5 on weekdays and T3 on weekends. The time intervals of 8:00–10:00 and 16:00–18:00 were the most crowded in X5, and 10:00–12:00, 12:00–14:00, and 14:00–16:00 were the most overcrowded in T3 (Figure 9c,d).

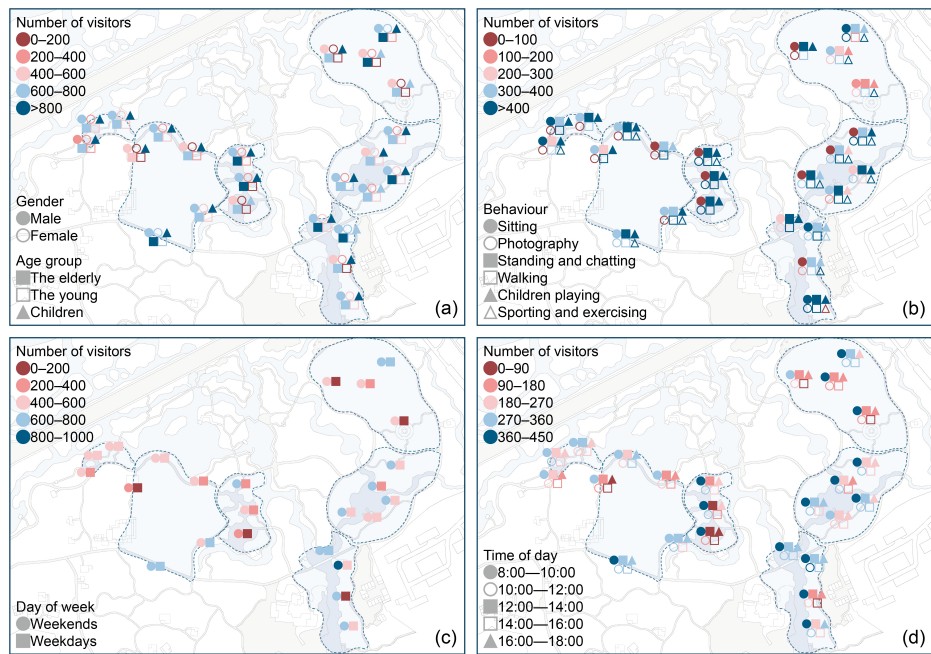

**Figure 9.** Mapping of visitors' physical activity and spatial behaviors. (**a**) Mapping of gender and age group of visitors. (**b**) Mapping of behavior of visitors. (**c**) Mapping of the day of the week of visitors. (**d**) Mapping of time of day of visitors.

*3.3. Environmental Factors Associated with Landscape Preferences*

Table 3 shows the results of Spearman's correlation analysis between environmental indicators and visitors' landscape preferences in Taiziwan Park. Nine of sixteen environmental indicators were significantly and positively correlated with visitors' landscape preferences, with higher values of the standardized coefficients indicating stronger correlations, including accessibility, VGI, soundscape quality, flowering plant abundance, adequacy of leisure facilities, tree abundance, thermal comfort, color richness, and building abundance. Of these indicators, accessibility and VGI had the highest correlation with visitors' landscape preferences. In addition, there are fewer indicators related to the landscape preferences of children visitors, and more indicators related to young visitors. More indicators of correlation indicate that this group is more susceptible to various environmental factors.

In addition, to compare the priority of importance between environmental indicators and to evaluate the degree of landscape preference of each sample point in an integrated manner, a PCA was conducted. In the PCA, the raw data were standardized to eliminate the negative effects of environmental factors due to differences in magnitude. Furthermore, a Kaiser–Meyer–Olkin test and Bartlett's spherical test were performed on the data to determine if the data were suitable for PCA. According to the results of the Kaiser–Meyer–Olkin test and Bartlett's spherical test (Table 4), the Kaiser–Meyer–Olkin was 0.63, which was greater than 0.6, and the significance was less than 0.05, indicating that the data supported the PCA. Meanwhile, according to the principle that the eigenvalue is greater than 1, four common factors were extracted, and the cumulative variance contribution was 82.134%. Thus, by extracting four common factors, 82.134% of the variance of the original variables were reflected (Table 5).

**Table 3.** The results of correlation analysis between environmental indicators and visitors' landscape preferences.

| Environmental Indicators | Visitor | Male | Female | The Elderly | The Young | Children |
|---|---|---|---|---|---|---|
| Accessibility | 0.972 *** | 0.906 *** | 0.933 *** | 0.494 * | 0.931 *** | 0.497 * |
| Visible green index | 0.921 *** | 0.898 *** | 0.875 *** | 0.513 ** | 0.835 *** | 0.513 * |
| Soundscape quality | 0.673 *** | 0.590 ** | 0.694 *** | 0.271 | 0.639 ** | 0.490 * |
| Flowering plant abundance | 0.670 *** | 0.557 ** | 0.692 *** | 0.497 * | 0.654 ** | 0.206 |
| Adequacy of leisure facilities | 0.611 ** | 0.626 ** | 0.537 ** | 0.094 | 0.569 ** | 0.638 ** |
| Tree abundance | 0.535 ** | 0.529 ** | 0.477 * | 0.312 | 0.427 * | 0.394 |
| Thermal comfort | 0.529 ** | 0.548 ** | 0.483 * | 0.510 * | 0.457 * | 0.065 |
| Color richness | 0.476 * | 0.340 | 0.523 ** | 0.619 ** | 0.438 * | −0.179 |
| Building abundance | 0.421 * | 0.480 * | 0.397 | −0.074 | 0.541 ** | 0.329 |

Note: * $p < 0.05$, ** $p < 0.01$, *** $p < 0.001$.

**Table 4.** Kaiser–Meyer–Olkin and Bartlett's spherical test.

| Indicators | Value |
|---|---|
| Sufficiently sampled Kaiser–Meyer–Olkin measure | 0.630 |
| Bartlett's spherical test of sphericity approximate chi-square | 96.675 |
| Degrees of freedom | 36.000 |
| Significance | 0.000 |

**Table 5.** Total variance explained.

| Component | Initial Eigenvalues | | | Extracted Sum of Squares Loadings | | | Rotated Sum of Squares Loadings | | |
|---|---|---|---|---|---|---|---|---|---|
| | Total | Variance % | Cumulative % | Total | Variance % | Cumulative % | Total | Variance % | Cumulative % |
| 1 | 3.451 | 38.347 | 38.347 | 3.451 | 38.347 | 38.347 | 2.581 | 28.678 | 28.678 |
| 2 | 1.617 | 17.969 | 56.317 | 1.617 | 17.969 | 56.317 | 1.798 | 19.978 | 19.978 |
| 3 | 1.318 | 14.641 | 70.958 | 1.318 | 14.641 | 70.958 | 1.537 | 17.073 | 17.073 |
| 4 | 1.006 | 11.176 | 82.134 | 1.006 | 11.176 | 82.134 | 1.476 | 16.404 | 16.404 |
| 5 | 0.721 | 8.015 | 90.149 | | | | | | |
| 6 | 0.368 | 4.089 | 94.237 | | | | | | |
| 7 | 0.302 | 3.356 | 97.593 | | | | | | |
| 8 | 0.138 | 1.529 | 99.122 | | | | | | |
| 9 | 0.079 | 0.878 | 100.000 | | | | | | |

Finally, the indicators were normalized using the variance contribution of the principal components as weights, and the weights of the indicators were calculated using PCA.

The calculation steps were as follows.

First, the coefficients of the linear combination were calculated using the following formula:

$$U_i^j = \frac{F_i^j}{\mu^j} \tag{1}$$

where $U_i^j$ is the coefficient in the linear combination corresponding to the $j$ component of the $i$ indicator, $F_i^j$ is the component matrix value corresponding to the $j$ component of the $i$ indicator, and $\mu^j$ is the square root of the eigenvalue of the $j$ component; $i = 1, 2, 3... 9$, $j = 1, 2, 3, 4$. The results of the calculations are as follows:

$$F_1 = 0.346X_1 + 0.259X_2 + 0.002X_3 + 0.489X_4 + 0.340X_5 + 0.474X_6 + 0.254X_7 + 0.350X_8 + 0.219X_9 \tag{2}$$

$$F_2 = 0.401X_1 + 0.326X_2 + 0.463X_3 - 0.087X_4 - 0.003X_5 - 0.079X_6 - 0.410X_7 - 0.384X_8 + 0.435X_9 \tag{3}$$

$$F_3 = 0.322X_1 - 0.120X_2 + 0.617X_3 - 0.065X_4 - 0.369X_5 - 0.138X_6 + 0.470X_7 + 0.226X_8 - 0.261X_9 \tag{4}$$

$$F_4 = -0.006X_1 + 0.621X_2 + 0.012X_3 - 0.048X_4 + 0.406X_5 - 0.263X_6 + 0.183X_7 - 0.216X_8 - 0.545X_9 \tag{5}$$

In the above formula, $X_1$, $X_2$,... $X_9$ are the nine related indicators in Table 6.

**Table 6.** Results of linear combination coefficients and weights.

| Indicator Number | Indicators | Principal Component 1 | Principal Component 2 | Principal Component 3 | Principal Component 4 | Combined Score Coefficient | Weight |
|---|---|---|---|---|---|---|---|
| - | Initial Eigenvalue | 3.451 | 1.617 | 1.318 | 1.006 | - | - |
| - | Explanation of variance | 38.35% | 17.97% | 14.64% | 11.18% | - | - |
| $X_1$ | Accessibility | 0.642 | 0.51 | 0.37 | −0.006 | 0.306 | 19.58% |
| $X_2$ | Visible green index | 0.482 | 0.414 | −0.138 | 0.623 | 0.255 | 16.36% |
| $X_3$ | Flowering plant abundance | 0.003 | 0.589 | 0.708 | 0.012 | 0.214 | 13.68% |
| $X_4$ | Adequacy of leisure facilities | 0.908 | −0.111 | −0.075 | −0.048 | 0.191 | 12.23% |
| $X_5$ | Soundscape quality | 0.631 | −0.004 | −0.424 | 0.407 | 0.147 | 9.43% |
| $X_6$ | Thermal comfort | 0.88 | −0.1 | −0.159 | −0.264 | 0.143 | 9.19% |
| $X_7$ | Tree abundance | 0.471 | −0.521 | 0.54 | 0.184 | 0.138 | 8.81% |
| $X_8$ | Color richness | 0.651 | −0.488 | 0.26 | −0.217 | 0.091 | 5.80% |
| $X_9$ | Building abundance | 0.407 | 0.553 | −0.3 | −0.547 | 0.077 | 4.91% |

Subsequently, the coefficients in the comprehensive score model were calculated from the coefficients in the linear combination as well as the variance of the principal component using the following formula:

$$Q_i = \frac{U_i^1 * R^1 + U_i^2 * R^2 + U_i^3 * R^3 + U_i^4 * R^4}{R^1 + R^2 + R^3 + R^4} \tag{6}$$

where $Q_i$ is the coefficient in the comprehensive score model; $U_i^1$, $U_i^2$, $U_i^3$, and $U_i^4$ are the coefficients of indicator i in the linear combination of components 1, 2, 3, and 4, respectively; $R^1$, $R^2$, $R^3$, and $R^4$ are the variances of the first, second, third, and fourth principal components, respectively. The model of the total score was as follows:

$$Y = -0.306X_1 + 0.225X_2 + 0.214X_3 + 191X_4 + 0.147X_5 + 0.143X_6 + 0.138X_7 + 0.091X_8 + 0.077X_9 \tag{7}$$

Finally, the indicators were normalized, and the weights ($W_i$) attributed to each factor were calculated using the following equation:

$$W_i = \frac{Q_i}{\sum Q_i} \tag{8}$$

The calculation formula of the *LPI* was as follows:

$$LPI = \sum_{i=1}^{n} w_i y_i \tag{9}$$

where *LPI* is the degree of visitors' landscape preference in urban parks, $w_i$ is the weight of the i indicator, and $y_i$ is the standardized value of the i indicator. The results of the PCA were obtained by estimating Equation (9) using the correlation data of the sample points.

The environmental indicator value of each selected point was filled in in Formula (7), and the preference score for each selection was calculated. Among the environmental indicators, accessibility had the highest weighting, followed by VGI, flowering plant

abundance, adequacy of leisure facilities, and soundscape quality. From Figure 10, the six selected points preferred by visitors were X5, T3, T2, T1, W4, and X3, which are mainly located at Xiaoyao Hillside, Tianyuan Terrace, and Wangshan Lawn, respectively.

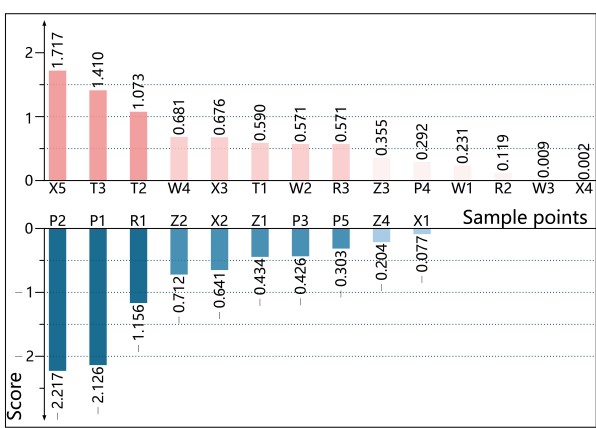

**Figure 10.** Landscape preference scores of sample points.

## 4. Discussion

### 4.1. The Relationships between Visitor Characteristics and Landscape Space

In summary, the study identified several characteristics of visitors to the Taiziwan Park Spring.

1.  Visitors to the park were mainly female. Compared to the other groups, female youths preferred to enjoy flowers and reenact social trends, which coincided with the characteristics of Taiziwan Park. With the tulip flower show and cherry blossoms as its main features in spring, Taiziwan Park attracted many visitors to enjoy the flowers and create online content following trends. The more visitors that came to the park to enjoy the flowers and post on social media, the more visitors were attracted to the park.
2.  The number of visitors peaked in the afternoon. Compared to other urban parks, visitors to Taiziwan Park were generally younger, and the young generally went to parks in the afternoon. However, the area around Taiziwan Park was dominated by scenic spots, and it took longer for urban residents to reach the park.
3.  The main behaviors of the visitors were sitting and photography. In addition to many tulips and cherry blossoms in the park, there was also a large lawn area for visitors to play and rest, and visitors often laid picnic mats on the lawn or sat on the ground. To cater for the tulip show, the park also had food stalls, cultural and creative stalls, and many seats to meet visitors' needs. The rich landscape of Taiziwan Park also catered to the needs of different visitors who used the park as a backdrop for portrait and landscape photography.

### 4.2. Spatiotemporal Distribution Characteristics of Visitors

In the spatiotemporal distribution characteristics of visitors, the distribution of visitors varied considerably among the various constituencies in the park. Xiaoyao Hillside, as the main viewing area for the flower show, attracted a significant number of visitors with its cherry blossoms and different varieties of tulips. Still, fewer leisure facilities were available on the site, and the overall stopping space was small, which led to traffic jams and disrupted visitors' activities due to photography on the road. Xiaoyao Hillside, however, had a low level of landscape preference. In contrast, Rose Garden had a smaller number of tulips, but a larger staying space, so many visitors were involved in photography there.

The Wangshan Lawn, with sizable accessible lawn areas, had a wide range of outdoor activities, such as walking, standing, chatting, and children playing. The sample points of the Wangshan Lawn were popular with male and child visitors. Similarly, Pipa Islet had a

large accessible lawn area and was close to the lake, so a large number of visitors sat and took photographs there. In contrast, there was only a single visitor behavior, due to the fact that the lawn area of Xiaoyao Hillside was largely inaccessible.

Tianyuan Terrace had plenty of food stalls and seats among tall trees, which met the needs of sitting and thermal comfort. Therefore, a large number of visitors gathered in this area. In contrast, Zhulianbi Waterfall also had tall trees and provided shade space, but with fewer leisure facilities. Consequently, most visitors would sit in Tianyuan Terrace rather than at the Zhulianbi Waterfall.

### 4.3. Effects of Environmental Factors on Landscape Preferences

It was found that visitors' landscape preferences were influenced by a combination of environmental factors, and there was a correlation between these factors. The study showed that accessibility and VGI were the most relevant to landscape preferences of visitors and accounted for the greatest weight in the PCA. The findings from this study further support the conclusions of previous studies, which mentioned that visitors' preferences were mainly driven by accessibility and physical distance to access of green space [19,20]. This phenomenon may be explained by the fact that a site with good accessibility creates convenient visiting conditions and a sense of security, making it easier for visitors to get to the site. Accessibility also satisfies visitors' preferences for activities, such as walking. Additionally, the high green coverage of urban parks, as suggested by most studies, plays a positive role in attracting visitors [21,22]. Vegetation, as a natural landscape component, helps reduce stress and restores attention, as well as a wide range of ecological benefits to the quality of life, including pollution and heat island mitigation and shade provision. The ecological problems caused by China's rapid development, particularly air pollution, have intensified the demand for vegetation in urban green spaces [23]. Higher vegetation coverage can more efficiently purify and increase negative oxygen ion concentrations in the air [24,25], thereby improving the micro-climate [26].

The findings of this study suggest that future urban greenspace landscape design should prioritize accessibility and VGI. Landscape planners should consider a combination of flowers and trees to enhance the attractiveness of the park landscape. Trees are more likely to meet visitors' landscape preferences than shrubs. With global climate change, creating a good continuum of shaded spaces will not only meet visitors' viewing needs but also their need for thermal comfort. The innovative integration of urban park visitor vegetation preferences into management and planning may ultimately contribute to enhancing visitor experiences and maximizing the many benefits of urban parks for both society and the environment [27].

Environmental indicators such as the adequacy of leisure facilities, flowering plant abundance, thermal comfort, soundscape quality, and tree abundance also influenced visitors' landscape preferences.

For adequate leisure facilities, visitors prefer places with seats, especially those with backrests and armrests [22]. Woolley [28] stated that the number of seats is closely related to the frequency with which a space is visited. Steps and stands also provide opportunities for people to sit, which can play a role in physical and psychological recovery. For flowering plant and tree abundance, vegetation, such as trees and flowers, contributes to creating restorative environments that reduce stress and mental fatigue, as well as maintaining and restoring the ability to direct attention [29], and has a positive effect on preferences [30]. Moreover, thermal comfort influenced visitors' landscape preferences, which were linked to indicators such as VGI and tree abundance. The people of China usually do not like to be exposed to sunlight for long periods; therefore, visitors prefer a landscape space with good thermal comfort with large, shaded trees. However, owing to the favorable climatic conditions in spring, the influence of thermal comfort on landscape preference will be relatively less weighted in this study. For soundscape quality, excessive noise can be physiologically uncomfortable for visitors, so automobiles, airplane noise, and human

voices are still not congruent with visitor expectations for natural settings and therefore reduce visitors' preference for the landscape [31].

*4.4. Suggestions for Park Landscape Improvement*

According to the characteristics of visitors in the park, this study made several recommendations.

1.  The flowers received great reception from female youths, so their needs should be considered, especially during spring. For example, park designers should provide an adequate number of additional mobile toilets to avoid long queues at the women's bathrooms; they should set up a sufficient number of photo-taking points to ease the pressure of visiting the park and reduce traffic congestion. At the same time, park designers should also consider the needs of other people; for instance, they should meet the demand of children in science by setting up corresponding introductory signs for the species of flower on display with quick response codes and audio.
2.  Urban parks should define peak crowd zones and organize crowds rationally according to their capacity. Although Taiziwan Park carried out a reservation system during spring, the overall flow of visitors was large. Traffic congestion tended to occur at critical points, increasing the pressure of visiting the park. To meet visitors' needs, the traffic tour route should be reasonably organized, with more prominent signposts at road turning points and entrances and exits dispersed to ensure one-way tours as far as possible.
3.  The main behaviors of visitors were photographing and sitting, so more leisure facilities should be set up to meet the needs of sitting during spring. Beyond that, some temporary stopping points, such as viewing platforms, should be set up in places with beautiful scenery to meet the needs of photographing.

In view of the spatial and temporal distribution of visitors in the park, it is suggested that for the long-term development of urban parks, regular flower shows create a distinctive vegetation landscape that can attract many visitors; however, during periods of high pedestrian traffic, corresponding leisure facilities should be set up to meet the needs of resting. For instance, more staying points should be set up in Xiaoyao Hillside to meet the needs of photography, and flowers should be planted in scenic spots to allow the flow of people and reduce foot traffic. Urban parks should also rationalize the flow of visitors according to their capacity and organize tour routes to avoid congestion during peak periods.

From the impact of environmental factors on landscape preferences, when managing and designing urban parks, multiple factors must be considered in an integrated manner to achieve optimal results. For different landscape spaces, landscape planners should consider the strengths and weaknesses of the site and tailor landscape effects accordingly. In areas with good accessibility, ornamental plants should be reasonably planted, and certain resting facilities should be set up to meet the needs of walking and sitting. In areas with high green coverage, stopping points should be provided to meet the needs of standing, chatting, sporting, and exercising individuals. In areas rich in ornamental plants, well-planned designated photo-taking points should be established to meet the needs of photography.

*4.5. Limitations and Future Studies*

Exploring the relationship between visitors' landscape preferences and urban parks is of great significance for improving park satisfaction, people's happiness, and urban sustainability [32,33]. The findings from this study showed that the environmental factors of Taiziwan Park, including accessibility, VGI, flowering plant and tree abundance, adequacy of leisure facilities, and thermal comfort, were differentially correlated with the landscape preferences of visitors in spring. These findings have implications for perfecting the landscape composition and configuration of parks to improve visitors' satisfaction with landscape space. Therefore, visitors' preferences should be taken into account in planning and management to create a beautiful, functional, and practical urban park [34].

This study has several limitations. In the first instance, although the use of UAV observations combined with ground observations to record the spatiotemporal distribution characteristics of visitors can provide access to large amounts of data in the short term, characteristics such as the occupational composition and usual place of residence cannot be understood through visual inspection and need to be elicited with questionnaires or interviews. In addition, the study time was only two months during the flower show, and other seasonal conditions should also be considered in subsequent study. Finally, the correlation analysis and PCA only explore the correlation between environmental factors and visitors' preferences and do not explore other relationship types. In response to these three limitations, future research should further integrate multi-source data and multiple methods in different study areas to understand the landscape preferences of different visitors. For example, The introduction of big data in field research enhances the comprehensiveness and adequacy of research objects. Extending the duration of the study can expand the generalization of the results in other seasons and verify the science and universality of the study. Finally, future studies should further analyze the correlation between various environmental factors and visitors' preferences and use various analytical methods to reveal the internal relationship between man and nature[35].

## 5. Conclusions

This study used fixed-point photography combined with mobile measurements to quantify the environmental factors of urban parks, UAV observations combined with ground observations to quantify visitors' landscape preferences, and correlation analysis to explore the environmental factors that influence visitors' landscape preferences. This is a useful method for understanding and improving the urban park environment and exploring the relationship between people and nature from a microscopic perspective. Using Taiziwan Park as an example, this study investigated and analyzed six sample areas and 24 sample points in the park during spring, resulting in the following main findings.

1. This study shows that visitors' landscape preferences are influenced by a combination of multiple environmental factors, including accessibility and VGI.
2. There were significant differences in landscape preferences between people of different genders and age groups. The main behaviors of visitors were siting and photography, and visitors were mainly located in areas with good vegetation landscapes and adequate leisure facilities. Furthermore, female visitors preferred to take photographs, whereas male visitors preferred to sit. Female visitors and the elderly were mostly found in Xiaoyao Hillside, while male visitors and children were found in Wangshan Lawn.
3. Accessibility and VGI were the most relevant to visitors' preferences, followed by flowering plant abundance, adequacy of leisure facilities, soundscape quality, thermal comfort, tree abundance, color richness, and building abundance.

The landscape space composition and environmental factors of urban parks are of great significance to visitors' landscape preference. These findings could help urban park planners and managers create more popular landscape spaces for visitors.

**Author Contributions:** Conceptualization, M.Y. and H.Y.; methodology, M.Y.; software, X.N. and M.Y.; validation, T.D., Y.L. and R.W.; formal analysis, M.Y.; investigation, M.Y.; resources, Z.B.; data curation, M.Y.; writing—original draft preparation, M.Y.; writing—review and editing, M.Y., H.Y. and X.N.; visualization, X.N.; supervision, R.W.; project administration, Z.B.; funding acquisition, H.Y. All authors have read and agreed to the published version of the manuscript.

**Funding:** This research was funded by the Zhejiang Provincial Natural Science Foundation of China (Grant No. LGF21E080001) and the National Natural Science Foundation of China (Grant No. 51508515).

**Data Availability Statement:** Not applicable.

**Acknowledgments:** We gratefully thank Shili Chen for helping us to process the data and express our appreciation to Han Wang and Shudan Liu from Zhejiang Agriculture and Forestry University for their invaluable assistance to the field research.

**Conflicts of Interest:** The authors declare no conflict of interest.

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
