# Peer review of "Effects of Urban Park Environmental Factors on Landscape Preference Based on Spatiotemporal Distribution Characteristics of Visitors"

_forests, doi:10.3390/f14081559_

Round 1

Reviewer 1 Report

Page 5 Table 2 – why not left indent indicator and description columns versus center.. looks odd

Table 3 pg.       6 – SAME COMMENT AS ABOVE

Tables 4 & 5 – and the text with the formulas – at this day and age with some many studies that use factor analysis…. Not sure if Tables 4 especially and all the formula information as well as perhaps Table 5…. And the formulas is actually necessary.  We all use a statistical package that runs these analyses and the forumals and calcultations aren’t really necessary to justify the results as we trust in those results.  I think that part of the paper can be reduced and report the results without the data tables and deletedthe formulas… as just takes up space..suppose it depends on the journal…but that’s my take on it. 

 Claims made early on that many visualization studies rely on one variable in photos... not so sure this is true because of the following series of papers that had at least 6 variables manipulated in the photos...with several panels of photos for the same location but the 6 variables altered both physical and social indicators - see 

Wynveen, C., Arnberger, A., Schneider, I.E., Cottrell, S.P., & Von Ruschkowski, E. (2020). Integrating place attachment into management frameworks: Exploring place attachment across the Recreation Opportunity Spectrum. Environmental Management. https://link.springer.com/article/10.1007/s00267-020-01292-7.

I appreciate the rigor of the analysis using multiple aspects via the mapping, multivariate and bivariate analyses.

Author Response

Response to Reviewer 1 Comments

Point 1: Page 5 Table 2 – why not left indent indicator and description columns versus center.. looks odd

Table 3 pg.6 – SAME COMMENT AS ABOVE

Response 1: The text format of the form will be modified according to the opinions of reviewer. Thank you for your comments.

Point 2: Tables 4 & 5 – and the text with the formulas – at this day and age with some many studies that use factor analysis…. Not sure if Tables 4 especially and all the formula information as well as perhaps Table 5…. And the formulas is actually necessary.  We all use a statistical package that runs these analyses and the forumals and calcultations aren’t really necessary to justify the results as we trust in those results.  I think that part of the paper can be reduced and report the results without the data tables and deletedthe formulas… as just takes up space..suppose it depends on the journal…but that’s my take on it. 

Response 2: Researchers familiar with factor analysis may be more aware of the overall analysis process and calculation methods, and these tables and formulas may be slightly redundant. However, researchers unfamiliar with it may need to understand the entire analysis process. Articles are retained for the time being and can be deleted if necessary. Thank you for your comments.

Point 3: Claims made early on that many visualization studies rely on one variable in photos... not so sure this is true because of the following series of papers that had at least 6 variables manipulated in the photos...with several panels of photos for the same location but the 6 variables altered both physical and social indicators - see 

 Wynveen, C., Arnberger, A., Schneider, I.E., Cottrell, S.P., & Von Ruschkowski, E. (2020). Integrating place attachment into management frameworks: Exploring place attachment across the Recreation Opportunity Spectrum. Environmental Management. https://link.springer.com/article/10.1007/s00267-020-01292-7.

Response 3: Thank you for your comments, the article has been modified to: Most of these studies have only modified a small number of variables for each group of control photos, making it difficult to study changes in landscape preferences in the case of multiple environmental factors.

Point 4: I appreciate the rigor of the analysis using multiple aspects via the mapping, multivariate and bivariate analyses.

Response 4: Thanks for your comments, we will continue to revise the article until it can be published in this journal. On behalf of my co-authors, we would like to express our great appreciation to you.

Reviewer 2 Report

Forests 2488503

Effects of urban park environmental factors on landscape preference based on spatiotemporal Distribution characteristics of visitors.

This is generally a well written research article but this reviewer has several suggestions that would improve the article.

Introduction

Authors need to build a stronger case why they used multisensory sampling. They stress the limitations of previous preference studies using only single physical landscape features, but do not build a case specifically for multiple sensory sampling.

Materials and Methods

Line 106- authors need to explain how the sixteen physical factors were selected for assessment. Is this based on previous studies or multisensory assessment literature/

Line 137- So the March to April sampling was done during optimum flowering conditions? This should be justified as opposed to sampling during other park usage seasons.

Results are well presented.

Discussion

I would recommend separating out park design recommendations into a separate section.

Line 391- Limitations and future studies

It should be stated that sampling during maximum vegetative flowering conditions during two months in the spring limits the generalizability of study results to the spring park conditions and that sampling during other seasons would enhance study findings.

Author Response

Response to Reviewer 2 Comments

Point 1: Authors need to build a stronger case why they used multisensory sampling. They stress the limitations of previous preference studies using only single physical landscape features, but do not build a case specifically for multiple sensory sampling.

Response 1: Thank you for your comments, the article has been added to the case, now changed to: In practice, the public does not evaluate environmental factors individually but evaluates the combination of environmental factors, resulting in preferences. Among the researches on multiple influencing factors of park landscape preference, most of the research focuses on visual scale. However, big data surveys show that in addition to visual factors, accessibility, thermal comfort, leisure facilities and other factors will have an impact on tourists' landscape preferences. Therefore, a systematic and comprehensive understanding of the relationship between visitor preferences and park environmental factors can provide management guidelines for landscape planners to optimize the landscape configuration of urban parks and promote greater integration of functional and practical urban parks into urban life.

Point 2: Line 106- authors need to explain how the sixteen physical factors were selected for assessment. Is this based on previous studies or multisensory assessment literature.

Response 2: The selection of 16 environmental factors is based on the synthesis of research and multi-sensory literature. Through random interviews with park visitors and reference to relevant literature, 16 environmental factors affecting visitors' landscape preference are determined, and the main factors affecting visitors' landscape preference are explored.

Point 3: Line 137- So the March to April sampling was done during optimum flowering conditions? This should be justified as opposed to sampling during other park usage seasons.

Response 3: The tulip exhibition period in Hangzhou Taiziwan Park is from March to April, and the overall landscape is beautiful and there is a large number of visitors. Therefore, this study chooses this period to explore the relationship between visitors' preference and landscape environment.

Point 4: I would recommend separating out park design recommendations into a separate section.

Response 4: Thanks for your comments, this part of the content has been modified, see the article for details.

Point 5: Line 391- Limitations and future studies

It should be stated that sampling during maximum vegetative flowering conditions during two months in the spring limits the generalizability of study results to the spring park conditions and that sampling during other seasons would enhance study findings.

Response 5: We will add and revise in the article and thank you for your comments.

Point 6: Results are well presented.

Response 6: Thanks for your comments, we will continue to revise the article until it can be published in this journal. On behalf of my co-authors, we would like to express our great appreciation to you.